# Aromatic Molecular Compatibility Attenuates Influenza Virus-Induced Acute Lung Injury via the Lung–Gut Axis and Lipid Droplet Modulation

**DOI:** 10.3390/ph18040468

**Published:** 2025-03-26

**Authors:** Yi Li, Jiakang Jiao, Haoyi Qiao, Conghui Wang, Linze Li, Fengyu Jin, Danni Ye, Yawen Chen, Qi Zhang, Min Li, Zhongpeng Zhao, Jianjun Zhang, Linyuan Wang

**Affiliations:** 1School of Chinese Materia Medica, Beijing University of Chinese Medicine, Beijing 102488, China; ly458945090@163.com (Y.L.);; 2School of Traditional Chinese Medicine, Beijing University of Chinese Medicine, Beijing 102401, China; 3State Key Laboratory of Pathogen and Biosecurity, Beijing Institute of Microbiology and Epidemiology, Academy of Military Medical Sciences, Beijing 100071, China; 4Beijing Minhai Biotechnology Co., Ltd., Beijing 102600, China

**Keywords:** acute lung injury, influenza A virus, volatile compounds, molecular compatibility, NLRP3 inflammasome, lipid droplets

## Abstract

**Background:** Acute lung injury (ALI) is a major cause of death in patients with various viral pneumonias. Our team previously identified four volatile compounds from aromatic Chinese medicines. Based on molecular compatibility theory, we defined their combination as aromatic molecular compatibility (AC), though its therapeutic effects and underlying mechanisms remain unclear. **Methods:** This study used influenza A virus (IAV) A/PR/8/34 to construct cell and mouse models of ALI to explore AC’s protective effects against viral infection. The therapeutic effect of AC was verified by evaluating the antiviral efficacy in the mouse models, including improvements in their lung and colon inflammation, oxidative stress, and the suppression of the NLRP3 inflammasome. In addition, 16S rDNA and lipid metabolomics were used to analyze the potential therapeutic mechanisms of AC. **Results:** Our in vitro and in vivo studies demonstrated that AC increased the survival of the IAV-infected cells and mice, inhibited influenza virus replication and the expression of proinflammatory factors in the lung tissues, and ameliorated barrier damage in the colonic tissues. In addition, AC inhibited the expression of ROS and the NLRP3 inflammasome and improved the inflammatory cell infiltration into the lung tissues. Finally, AC effectively regulated intestinal flora disorders and lipid metabolism in the model mice, significantly reduced cholesterol and triglyceride expression, and thus reduced the abnormal accumulation of lipid droplets (LDs) after IAV infection. **Conclusions:** In this study, we demonstrated that AC could treat IAV-induced ALIs through multiple pathways, including antiviral and anti-inflammatory pathways and modulation of the intestinal flora and the accumulation of LDs.

## 1. Introduction

Zoonotic viral diseases have emerged and re-emerged throughout history and continue to pose a significant threat to global human health [1]. As coronavirus disease 2019 (COVID-19) continues to spread worldwide, it is likely to overlap with influenza virus infections during each influenza season, thus posing a significant health burden [2]. Given the significant morbidity and mortality associated with influenza pandemics each year and the potential for influenza to have a similar impact to that of COVID-19 [3,4], the development of drugs against influenza is necessary.

Viral invasion through the respiratory tract induces the overexpression of various inflammatory factors, which can result in a “cytokine storm” [5,6], a condition that represents the core of acute lung injury (ALI) and is one of the most significant mechanisms leading to functional organ damage in patients [6,7]. In the absence of prompt and appropriate treatment, an ALI is likely to progress to severe acute respiratory distress syndrome (ARDS) [8], as observed during the recent COVID-19 outbreak, in which ALI/ARDS was a significant cause of mortality in a substantial number of critically ill patients [9,10]. Therefore, the development of effective therapies to suppress virus-induced ALIs represents a promising strategy for the treatment of viral pneumonia.

In recent years, the emerging “gut–lung axis” hypothesis has posited that the mucosal tissues of the lungs and intestines serve as important barriers against the external environment and can influence each another’s immune responses through the systemic circulation [11,12,13]. It has been demonstrated that impaired intestinal barrier integrity due to intestinal flora dysbiosis may result in the migration of SARS-CoV-2 from the lungs to the intestinal lumen via the circulatory and lymphatic systems. This finding is consistent with the gastrointestinal symptoms observed in some patients with COVID-19 [14,15]. Furthermore, it has been posited that traditional Chinese medicine (TCM) may influence the composition of the intestinal flora [16,17] and its metabolism [18,19], thereby modulating the intestinal flora of patients with influenza or SARS-CoV-2; evidence on this mechanism could elucidate the mechanism of TCM treatment for lung disease.

Viruses are intracellular microorganisms that exploit the host’s metabolic processes to meet their biosynthetic needs. For example, the lipid metabolism within the host cell is hijacked for the purpose of supplying energy to the viral particles during the process of replication or assembly [20,21]. Lipid droplets (LDs) are sites for viral replication and assembly [22]. Consequently, the accumulation of cytoplasmic LDs is typically observed in virus-infected cells, while the upregulation in the early steps of LD formation, in turn hijacking autophagy, enables ZIKV to utilize the host’s lipid metabolism to drive viral replication [23,24]. It has been proposed that apolipoprotein E (ApoE) may act as a resistance factor against influenza A virus (IAV), with deficiency in ApoE expression leading to the accumulation of cholesterol in the cell membrane, thereby increasing IAV-associated salicylic acid receptor activity and subsequent viral invasion [25]. These studies have suggested that targeting the host’s lipid metabolism could be a potential strategy for antiviral therapy.

TCM is distinguished by its multi-level, multi-pathway, and multi-targeted approach to the prevention and treatment of infectious diseases [26,27]. Among the various therapeutic modalities employed in TCM, aromatic herbs have been demonstrated to possess efficacy and potential in the prevention and treatment of COVID-19 [28,29]. Volatile substances are the main active ingredients in aromatic Chinese medicines, and modern pharmacological studies have shown that volatile components may play a role in the prevention and treatment of COVID-19 by inhibiting cellular cytokine storms, alleviating lipid peroxidation damage, and regulating host immune function [30,31,32,33]. It has been found that patchouli alcohol in patchouli oil has anti-H2N2 influenza virus effects [34]. It was able to enhance the body’s immune response and attenuate systemic and pulmonary inflammatory responses [35]. Eucalyptol can regulate the body’s immune function and inhibit ALIs caused by cytokine storms [36,37,38]. Based on the properties of aromatic Chinese medicines [32], more research should be performed on their unique medicinal properties, chemical compositions, and mechanisms of action to provide possible treatment for viral pneumonia and to aid in the development of corresponding medicines. In view of the significant role of aromatic herbs in the treatment of viral pneumonia, our team screened four volatile compounds from aromatic Chinese medicines and, through TCM [39] and molecular compatibility theory [40], defined the synergistic application of these four compounds as aromatic molecular compatibility (AC) [41]. However, the effect and mechanism of AC in the treatment of influenza pneumonia are not clear.

In this study, we employed a variety of biological techniques to validate the therapeutic effects of AC on mice with IAV-induced ALIs and their mechanisms. First, the inhibitory effect of AC on influenza virus was evaluated using in vitro and in vivo experiments. Subsequently, based on the lung–gut axis theory, we investigated the effects of AC on inflammatory injury in the lung and colon tissues and the gut flora in the model mice. Meanwhile, the effect of AC on lipid metabolic remodeling was investigated by metabolomics and verified by lipid droplets. In conclusion, this study demonstrated that AC could provide multi-pathway treatment for IAV-induced ALIs through antiviral and anti-inflammatory effects and modulation of the intestinal flora and lipid metabolism disorders. These findings provide a new strategy for the clinical treatment of respiratory diseases using active ingredients derived from aromatic herbs.

## 2. Results

### 2.1. The Safety and Antiviral Activity of AC In Vitro

The in vitro safety and antiviral efficacy of the AC treatment were initially established by measuring the 50% cytotoxic concentration (CC50), the 50% cell effective concentration (IC50), and the selectivity index (SI). In the MDCK cell model, for example, the results showed that the CC50 (168.90 ± 3.09 μg/mL) of the AC treatment in normal cells and the SI (20.8) (Figure 1A–E) in the IAV infection model were significantly higher than those for patchouli alcohol, carvacrol, p-cymene, and eucalyptol (Appendix A), suggesting that the AC of the four monomers may realize toxicity-reducing and synergistic effects through the synergistic effect among the multiple components. The therapeutic effect of AC was weaker when compared to that of oseltamivir (Figure 1F). The higher the SI value, the more potent a drug is in terms of its antiviral efficacy. However, this value only represents the direct inhibitory effect of the drug on the virus at the cellular level, and it is necessary to evaluate the indirect antiviral effect on the body through other pathways to comprehensively and comparatively assess the true efficacy of a drug’s action. We speculated that AC might exert its protective effect through other pathways than just direct inhibition of the influenza virus. Therefore, we investigated the protective effect of AC against influenza pneumonia in a mouse model.

### 2.2. AC Protects Against Mice Infected with the Influenza Virus

The objective of this study was to investigate the effect of AC on the efficacy and survival of H1N1 mice. This study employed a mouse model infected intranasally with influenza A virus (IAV) PR8 strain (10^3^ TCID50, 50 μL). The AC combination was administered continuously for 14 days following infection (Figure 2A). As illustrated in Figure 2B, mortality began on day 6 in the vehicle-treated group compared with the controls and continued until day 9 post-infection. In contrast, treatment with the AC combination significantly enhanced the survival rate of the mice infected with H1N1 in a dose-dependent manner, with a survival rate of 30% in the low-dose group and 50% in the high-dose group. The survival rate of the H1N1 mice treated with oseltamivir was 60%, which was close to the effect of the high-dose AC treatment. Subsequently, the groups were treated with oseltamivir, low-dose AC treatment, and high-dose AC treatment within 4 days of influenza virus infection (Figure 2C). The body weights of the vehicle, oseltamivir, low-dose AC, and high-dose AC groups all exhibited a decreasing trend in response to influenza virus infection, reaching reductions of 80.16%, 90.53%, 83.10%, and 89.59%, respectively (Figure 2D). The results demonstrated that both the high dose of the AC combination and oseltamivir were effective at ameliorating the dramatic body weight loss in the H1N1-infected mice compared with that in the vehicle group. The inhibitory effect of AC on the influenza virus was assessed by determining the lung viral load in the H1N1-infected mice. Interestingly, both the high- and low-dose AC groups demonstrated a reduction in influenza virus replication in their lung tissues compared with that in the vehicle group (Figure 2E). However, there was still a difference in this regard between the AC groups and the oseltamivir group. Nevertheless, in terms of the lung index and lung index inhibition, the high-dose AC group exhibited effects comparable to those in the oseltamivir treatment group (Figure 2F,G), indicating that the protective effect of AC in the H1N1-infected mice may have been exerted through additional pathways in addition to direct viral inhibition.

### 2.3. AC Inhibits IAV-Induced Lung Inflammation in Mice with ALIs

Our subsequent investigations focused on the therapeutic efficacy of AC in alleviating IAV-induced ALIs, with a particular focus on the potential of AC to suppress inflammation in the H1N1-infected mice. The results demonstrated that the lung tissues of the mice in the vehicle group exhibited extensive inflammatory infiltration and severe damage to the alveolar structures in comparison to those of the control group. Furthermore, treatment with oseltamivir was found to be effective in ameliorating lung inflammation and alveolar structural damage. Furthermore, we observed that as the AC dose increased, the treatment effect became more pronounced, with partial restoration of the alveolar structures, a reduction in interstitial thickening and edema, and a reduction in hemorrhage and inflammatory cell infiltration (Figure 3A). Cytokine storms resulting from uncontrolled local inflammation are a significant contributing factor to mortality in numerous patients with respiratory viral pneumonia. In order to investigate this further, a study was conducted in order to examine the expression levels of cytokines after the AC treatment. As illustrated in Figure 3B–G, the levels of TNF-α, IL-6, IFN-γ, IP-10, MCP-1, and CCL3 were markedly elevated in the vehicle group, indicating that following infection with the influenza viruses, the lung tissues produced a plethora of proinflammatory mediators, thereby establishing an inflammatory milieu within the lungs. The administration of AC, particularly in the high-dose group, notably reduced the expression levels of TNF-α, IL-6, IFN-γ, IP-10, MCP-1, and CCL3 in the lung tissues of the mice. This finding suggests that the AC treatment was capable of suppressing cytokine storms in the lung tissues.

Subsequently, we investigated the effect of AC on the macrophage phenotype in the lung tissues of the H1N1-infected mice. Figure 3H shows that F4/80 and iNOS were significantly expressed in the lung macrophages in the vehicle group, indicating that the macrophages were highly polarized toward the M1 phenotype at the onset of ALI; moreover, the CD206 fluorescence was increased compared with that in the control group, suggesting that the macrophages also polarized toward the M2 phenotype at the onset of ALI to attenuate tissue injury in the vehicle group. The AC treatment, especially in the high-dose group, effectively reduced the intensity of iNOS fluorescence and increased the intensity of CD206 fluorescence, suggesting that high-dose AC treatment could promote the activation of the alveolar M2 macrophages, thereby inhibiting the inflammatory response and alleviating ALI.

### 2.4. AC Inhibits IAV-Induced Colonic Lesions in Mice with ALI

Numerous studies [11,12] have shown that different types of respiratory viruses cause lung damage, as well as intestinal inflammation. Therefore, we investigated the ameliorative effect of AC on the intestinal inflammation and barrier dysfunction in the H1N1-infected mice. Compared with these properties in the control group, the colons in the group with IAV-induced ALIs also exhibited significant inflammatory cell infiltration and crypt disruption, and these histopathological changes in the colons were ameliorated by the oseltamivir and AC treatments (Figure 4A). Moreover, the levels of the inflammatory indicators TNF-α, IL-6, IFN-γ, IP-10, MCP-1, and CCL3 were significantly elevated in the colons of the vehicle group mice. However, oseltamivir and the AC treatment (especially at high doses) significantly suppressed the expression of these inflammatory indicators (Figure 4B–G). These results suggest that AC has potent anti-inflammatory effects on the colonic tissues of mice with IAV-induced ALIs.

We then investigated the effect of AC on the colonic epithelial barrier, as inflammation is an important factor in intestinal epithelial barrier damage. Figure 4H shows that the expression of the intestinal tight junction proteins ZO-1 and occludin was significantly downregulated in the vehicle group, suggesting that the intestinal barrier was impaired in the H1N1-infected mice but that the AC treatment counteracted the trend in the downregulation of ZO-1 and occludin expression and thus effectively inhibited influenza virus-induced colonic epithelial barrier damage in the ALI mice. In conclusion, AC ameliorated intestinal inflammatory injury in the H1N1-infected mice, including IAV-induced inflammatory injury in the intestinal tissues.

### 2.5. AC Inhibits ROS and the NLRP3 Inflammasome in Mice with IAV-Induced ALIs

Oxidative stress is a common consequence of inflammatory responses and is associated with elevated ROS levels. These high ROS levels not only cause damage to the epithelial cells but also stimulate the assembly and activation of the NLRP3 inflammasome complex and enhance the overall immune response [42]. The present study demonstrated that the accumulation of ROS in the lungs was significantly elevated after influenza virus infection (Figure 5A,B). Furthermore, the administration of the AC treatment, particularly in the high-dose group, markedly inhibited the increase in ROS and restored them to normal levels. In parallel, the protein expression of NLRP3, ASC, and caspase-1 was significantly upregulated in the lungs following viral invasion (Figure 5C,D). However, this trend was reversed in the high-dose AC treatment group, where the NLRP3 inflammasome was significantly downregulated in conjunction with IL-1β (Figure 5E). Overall, AC can exert anti-inflammatory effects and ameliorate IAV-induced ALIs by inhibiting the expression of ROS, thereby suppressing the activation of the NLRP3/ASC/caspase-1 pathway.

### 2.6. AC Suppresses Immune Cell Infiltration in Mice with IAV-Induced ALIs

In patients with influenza virus and other respiratory viral infections, the massive infiltration of immune cells and the consequent secretion of excessive proinflammatory cytokines, resulting in dysregulation of the immune system, can be observed. The results showed significant activation of the CD8^+^ T-cells in the peripheral blood of the mice after IAV infection, while there was a decrease in the number of CD4^+^ T-cells in the peripheral blood (Figure 6A–D), as well as cytokine storm production (Figure 6B–G), which was consistent with previous reports [43,44]. The AC treatment was effective in reversing this phenomenon in the peripheral blood and spleens (Appendix A) of the model mice.

Meanwhile, we assessed the immune cell infiltration in the mice’s lung tissue through immunohistochemistry. We found that the AC treatment, especially the high dose of the AC treatment, was able to effectively suppress the expression of CD8 and CD68 in the lung tissues (Figure 6E–H), indicating a reduction in the inflammatory infiltration of the T-cells and monocytes/macrophages into the lung tissues. Also, AC could modulate the infiltration of NK cells (CD56^+^) and B-cells (CD19^+^) into the lung tissues (Appendix A). In conclusion, AC suppressed the immune cell infiltration caused by influenza virus invasion, thereby alleviating influenza virus-induced immunopathological damage.

### 2.7. AC’s Modulation of the Intestinal Flora in Mice with IAV-Induced ALIs

To evaluate the effect of AC on the gut microbial composition of the mice with IAV-induced ALIs, we used 16S rDNA sequencing. According to the α-diversity analysis (Figure 7A), the high-dose AC treatment suppressed the Chao1 index in the H1N1-infected mice, with a concomitant increase in the Shannon index, suggesting that it could effectively reduce the IAV-induced disruption of the intestinal flora and restore it to normal levels. Although there was no significant difference between the AC treatment groups in terms of the Simpson and Pielou indices, the high-dose AC treatment group was also more similar to the control group in this way, suggesting that the number of species in the flora in this group was more uniform and stable. In the β-diversity analysis (Figure 7B), the PCA (PCA1 = 38.27%, PCA2 = 18.5%), PCoA (PCoA1 = 57.28%, PCoA2 = 10.64%), and MDS analysis (stress = 0.06) revealed significant separation between the different groups. Moreover, the high-dose AC group was widely distributed between the normal and model groups, suggesting that AC brought the microbiota of the H1N1-infected mice closer to the gut microbiota of the normal mice.

We observed differences in the overall composition of the gut microbiota when we examined the relative abundance of the top 20 phyla in the gut flora (Figure 7C). In general, the *Firmicutes/Bacteroidetes* (F/B) ratio, a common microbial marker associated with gut health (Figure 7D), was significantly lower in the H1N1-infected mice (0.8726 ± 0.1962) than that in the control mice (1.598 ± 0.2333) and tended to be greater in the AC-treated group than that in the normal mouse group (1.542 ± 0.2464). There were differences in the relative abundance of the top 30 genera in the gut flora composition (Figure 7E). Specifically, *Vibrio*, *Pseudoalteromonas*, *Agathobacter*, and *Erysipelatoclostridium* were significantly more abundant in the H1N1-infected mice than in the controls, while *Akkermansia*, *Dialister*, *Faecalibacterium*, and *Klebsiella* were significantly less abundant (Figure 7F,G). The AC treatment successfully reversed these changes. Based on the differences in the composition of the gut microorganisms in each group, we analyzed the functions of their differentially abundant flora. It is noteworthy that despite the relatively low enrichment in lipid metabolism (glycerol degradation), the differences observed were particularly pronounced (Figure 7H). This finding indicated that AC could protect the H1N1-infected mice by regulating their lipid metabolism, which needs to be verified through lipid metabolomics. In conclusion, it was demonstrated that AC could effectively regulate the structural composition of the intestinal flora in the mice with IAV-induced ALIs.

### 2.8. AC’s Regulation of the Lipid Metabolism in the Mice with IAV-Induced ALIs

In the context of the lipid metabolomics analysis, the level 3 KEGG analysis results for the top 20 lipid metabolite pathways revealed predominant enrichment in glycerophospholipid, glycerolipid, and cholesterol metabolism (Appendix A). The PCA and PLS-DA (Figure 8A–C) revealed disparate metabolic profiles among the different groups of samples. A total of 91 metabolites showed an upregulated abundance and 30 metabolites showed a downregulated abundance in the H1N1-infected mice compared to these values in the control mice (Figure 8D), of which 89 metabolites had an upregulated abundance and 36 metabolites had a downregulated abundance after treatment in the high-dose AC group (Figure 8E), with differentially abundant metabolites (Appendix A) enriched mainly in lysophosphatidylcholine, phosphatidylcholine, phosphatidylethanolamine, and TGs (Figure 8F). Significant modulation of glycerophospholipid metabolism, glycerolipid metabolism, and cholesterol metabolism after the AC treatment was found through KEGG annotation of the differentially abundant metabolites between groups (Figure 8G–I). Finally, we used Spearman’s correlation data constructs to establish 16S–metabolomic correlation analyses, which showed that TGs and phosphatidylcholine were negatively correlated with 22 flora taxa and positively correlated with 23 flora taxa. Phosphatidylethanolamine, lysophosphatidylcholine, and fatty acids were positively correlated with 22 flora taxa and negatively correlated with 23 flora taxa (Appendix A). Collectively, these findings suggest that the protective effect of AC on the H1N1-infected mice may have been achieved through the regulation of the cholesterol and TG levels in these mice.

### 2.9. AC Inhibits IAV-Induced LD Accumulation

LDs, which store neutral lipids such as TGs and cholesterol, play a pivotal role in the body’s innate immune response. To ascertain the regulation of the cholesterol and TG levels by AC, LDs were examined in this study. The results indicated the notable accumulation of LDs and elevated levels of cholesterol and TGs in the IAV-infected cells. The AC treatment resulted in a significant decrease in the fluorescence intensity of the LDs (Figure 9A,B) and a significant decrease in the cholesterol and TG levels in the cells (Figure 9C,D). In contrast to these properties in the H1N1-infected mice, the BODIPY fluorescence in the lung tissue was significantly decreased in the AC-treated mice (Figure 9E,F), and their cholesterol and TG levels were significantly elevated, suggesting that LDs were significantly elevated in the lung tissues of the IAV-infected mice with ALIs. Moreover, the AC treatment reversed this trend (Figure 9G,H). Finally, we observed a reduction in the content of the lipid transporter protein ApoE in the IAV-infected cells and mice. The AC treatment resulted in an increase in the ApoE concentration (Appendix A). In conclusion, AC was able to directly regulate lipid metabolism in the mice with IAV-induced ALIs and effectively inhibited the abnormal accumulation of cholesterol and TGs, as well as LDs, in the IAV-infected cells and mouse lung tissues, and this lipid-regulating mechanism may have been achieved through an effect on the expression of ApoE.

## 3. Discussion

The active ingredients in aromatic Chinese medicines that have shown beneficial effects in the prevention and treatment of COVID-19 have been elucidated [31]. The volatile oil components of aromatic Chinese medicines are predominantly small molecules with strong lipid solubility, which facilitates their absorption by organisms [32]. Modern pharmacological studies have demonstrated that the volatile oils in Chinese medicines and their monomers exhibit distinctive anti-inflammatory, antibacterial, and antiviral properties [30], with a wide range of applications [31,32] in the treatment of the respiratory, gastrointestinal, central nervous, and cardiovascular systems.

In a previous study, our team obtained aromatic molecular compatibility (patchouli alcohol: carvacrol: p-cymene: eucalyptol at 1:1:2:4, *w*/*w*) for the first time based on the theory of molecular compatibility [41]. Thus, this study attempted to investigate the efficacy and mechanism of action of AC further in the treatment of IAV-induced ALIs to identify the potential of AC as a therapeutic agent for ALIs.

In the present study, it was found that AC was protective in a cellular model of IAV infection (Figure 1), as well as in a mouse model (Figure 2B), in a dose-dependent manner within a defined safe range. Notably, while AC demonstrated less pronounced efficacy in directly inhibiting viral replication than that of oseltamivir, it exhibited a remarkable capacity to enhance the survival of and improve the lung index in the H1N1-infected mice (Figure 2). This leads us to hypothesize that the protective effect of AC on the H1N1-infected mice may not have solely been antiviral in nature but rather may have exerted therapeutic effects through alternative pathways. A clinical study demonstrated that lung injuries caused by SARS-CoV-2 and IAV were primarily due to an excessive inflammatory response rather than direct damage to the epithelium of the lungs by the virus. A cytokine storm, which is caused by an excessive immune response, may develop into ALI/ARDS, which can lead to respiratory failure or even death. Consequently, our subsequent research will concentrate on the potential therapeutic benefits of AC in the treatment of IAV-induced ALIs.

The lung–gut axis hypothesis posits that lung disease is associated with intestinal damage and disruption of the intestinal flora [13]. Consequently, the present study initially sought to assess the impact of AC on the lung and intestinal inflammatory damage in mice with IAV-induced ALIs. The results demonstrated that there was significant infiltration of the inflammatory cells into the lungs of the H1N1-infected mice (Figure 3A), accompanied by inflammatory damage in the colon (Figure 4A). This indicated that the lung and colon tissues were damaged to a considerable extent. The degree of inflammatory damage observed in the H1N1 mouse model varied. Treatment with AC effectively suppressed the inflammatory damage in the lungs and intestinal tissues and increased the protein expression of ZO-1 and occludin in the colon tissues (Figure 4H). The invasion and replication of the virus result in disruption of the immune balance, which in turn leads to the excessive and uncontrolled release of proinflammatory cytokines. The secretion of IFN and proinflammatory cytokines can rapidly recruit intrinsic immune cells to the site of infection and form a “positive feedback” loop under the stimulation of cytokines such as IFN-γ, TNF-α, and IL-6. This is accompanied by the release of large amounts of chemokines (CC and CXC chemokines) and an exacerbated inflammatory response. Following the AC treatment, the expression of TNF-α, IL-6, INF-γ, IP-10, CCL2, and CCL3 was significantly reduced in the lung (Figure 3) and colon tissues (Figure 4), effectively calming the cytokine storm. Concomitantly, the inflammatory response is accompanied by intense oxidative stress, which is capable of activating the NLRP3 inflammasome [42]. The NLRP3 inflammasome plays an important role in various types of ALIs by mediating the generation of inflammatory mediators and the infiltration of inflammatory cells, which promotes pulmonary edema and exacerbates lung tissue injury. In the present study, we observed that the AC treatment reduced the ROS levels and activated the NLRP3/ASC/caspase-1 pathway in the lung tissues of the H1N1-infected mice. This led to an improvement in the inflammatory microenvironment of the lung tissues (Figure 5).

When the virus invades lung tissue, the lung macrophages are overactivated and differentiate into M1 macrophages, releasing large amounts of proinflammatory factors that cause cytokine storms. The regulation of macrophage polarization is considered an effective strategy for inhibiting ALIs. The abundance of T-cells [45] is an important indicator of the body’s immune function, and the ratio of CD4^+^/CD8^+^ T-cells can reflect the body’s immune status [44]. In general, an increase in this value suggests the predominance of the positive regulation of the immune response, while a decrease in this value indicates that the body’s immune function is at the stage of exhaustion. The B-cells are producers of specific antibodies that not only play a role in antiviral and anti-infective effects but also protect the body from secondary infections. The AC treatment effectively reduced the iNOS expression and increased the CD206 expression in the lung tissues (Figure 3H), suggesting that AC could effectively inhibit the polarization of the M1 macrophages in the lungs, promote the differentiation of the M2 macrophages, regulate the polarization of the lung macrophages, and inhibit the progression of inflammation in the lungs. Furthermore, AC significantly reduced the proportion of CD8^+^ T-cells and increased the ratio of CD4^+^/CD8^+^ T-cells in the blood and lung and spleen tissues of the H1N1-infected mice (Figure 6). Additionally, AC regulated the expression of CD19^+^ and CD56^+^, thus affecting the ratio of B-cells and NK cells (Appendix A). This suggests that the direct antiviral effect of AC is not the only therapeutic pathway but that it also acts through indirect antiviral pathways (anti-inflammatory, immunomodulatory, or tissue-protective), which significantly improved the lung pathology in the model mice, and that this indirect pathway is the embodiment of the multi-targeting action of TCM.

As previously described [15], the intestinal flora, as part of the lung–gut axis, affects the progression of lung disease. Therefore, the present study investigated the effect of AC on the intestinal flora of the mice with IAV-induced ALIs. The results (Figure 7) demonstrated that the Chao1 and Shannon indices were increased in the H1N1-infected mice, indicating that the intestinal flora was disturbed by IAV. However, this effect was suppressed and stabilized after the AC treatment. According to the β-diversity analysis, PCA, PCoA, and MDS analysis, there was a significant separation between the healthy and H1N1-infected mice. Additionally, there was some overlap between the AC-treated group and the other two groups, with the AC-treated group exhibiting greater proximity to the intestinal microbiota of the normal mice. The application of AC resulted in a notable reduction in the population of pathogenic bacteria, including *Vibrio, Pseudoalteromonas, Agathobacter*, and *Erysipelatoclostridium*. Conversely, the abundance of probiotic bacteria, such as *Akkermansia, Dialister, Faecalibacterium*, and *Klebsiella*, significantly increased. Studies have shown that *Vibrio* spp. produce enterotoxins that cause metabolic acidosis and acute renal failure. *Pseudoalteromonas* has been associated with intestinal barrier dysfunction and pulmonary infections, which may lead to sepsis and multiple organ dysfunction syndrome. The presence of *Agathobacter* and *Erysipelatoclostridium*, which are characteristic flora of chemical peptic ulcers, suggests an association with inflammatory diseases. In contrast, *Akkermansia, Dialister, Faecalibacterium*, and *Klebsiella* can exert probiotic effects by modulating intestinal barrier function (mucus production and the immune system), mitochondrial activity, and lipid metabolism. Analyses of differentially abundant intestinal flora have demonstrated that although the enrichment of lipid metabolism is relatively low, the observed differences are particularly striking. Additionally, several studies have demonstrated that lipid metabolism is altered following viral infection. In our subsequent studies of lipid metabolism, 91 metabolites showed an upregulated abundance and 30 metabolites showed a downregulated abundance in the H1N1-infected mice (Figure 8). In the model mice, 89 metabolites had an upregulated abundance, and 36 metabolites had a downregulated abundance after the AC treatment. Conditional pathogenic bacteria (*Streptococcus* and *Citrobacter*) were positively correlated with TGs and PC, while probiotic bacteria (*Faecalibacterium* and *Butyricoccus*) were negatively correlated with LPC and TGs. This result suggests that IAV infection may drive lipid metabolism disorders and inflammatory cascade responses by disrupting the flora–metabolism interaction network. The AC treatment significantly modulated the glycerophospholipid, glycerolipid, and cholesterol metabolism pathways in the H1N1-infected mice, suggesting that the regulation of glycerol ester and cholesterol metabolism may be a potential mechanism for the treatment of influenza.

LDs are unilamellar organelles distributed in the cytoplasm and nucleus for the storage of neutral lipids such as TGs and cholesterol. Several findings [23] have suggested that LDs can serve as a platform for viral replication and assembly, as well as a source of lipids and energy for viral replication. For example, intestinal viruses [46] may actively take up external fatty acids and synthesize neutral lipids from fatty acids for temporary storage in lipid droplets, providing a source of lipids for the establishment of viral replication compartments. This suggests that targeting the metabolism of LDs may be a potential strategy for the treatment of viral pneumonia. Interestingly, the data on TG and cholesterol metabolism obtained from the lipid metabolism analysis in this study directly determined the aggregation status of the LDs in vivo. Therefore, we investigated the LDs accordingly. LDs significantly accumulated in the IAV-infected cells, as well as in the mouse lung tissues (Figure 9). This accumulation was accompanied by an increase in cholesterol and TG levels. Furthermore, the AC treatment effectively reduced the aggregation of LDs, resulting in a subsequent decrease in cholesterol and TG levels. ApoE is a lipid transporter protein that plays a role in the conversion and metabolism of lipoproteins. It has been demonstrated that ApoE facilitates the replication of hepatitis C virus [47]. Additionally, it has been demonstrated [25] that ApoE deficiency results in the accumulation of cellular cholesterol, which in turn leads to an increase in influenza virus salicylic acid receptors. These findings suggest that ApoE plays an important role in antiviral processes. In the present study, we observed a significant decrease in the ApoE expression in the IAV-infected cells and lung tissues. However, AC reversed this trend. In conclusion, these findings suggest that AC can be used to treat IAV-induced ALIs and improve survival by affecting the expression of the ApoE protein, thereby modulating lipid transport metabolism in organisms. This resulted in a decrease in the accumulation of LDs in the H1N1-infected mice and a reduction in the energy supply required for viral replication. This study has certain limitations. For instance, we did not correlate LD metabolism with the inflammatory network, nor did we validate the degradation of the TGs in the LDs. In a follow-up study, we will conduct a more in-depth investigation of the inflammatory microenvironment and the LD metabolism network by applying AC.

In summary, this study validated the protective effect of AC against IAV-induced ALIs through multiple pathways, including antiviral and anti-inflammatory pathways and modulation of the intestinal flora and lipid metabolism. This innovative molecular compatibility (Figure 10) allows for the exploration of potential TCM-based therapeutic approaches. Also, in terms of this molecular compatibility, the molecular compounds in TCM have well-defined compositions, precise mechanisms of action, and controllable quality, which is in line with modern medical standards and is expected to be strategic for the modernization of TCM.

## 4. Materials and Methods

### 4.1. Chemicals and Reagents

Patchouli alcohol (purity ≥ 98%), carvacrol (purity ≥ 99%), p-cymene (purity ≥ 98%), and eucalyptol (purity ≥ 99%) were purchased from Chengdu Herbpurify Co., Ltd. (Chengdu, China); the purity of all four compounds was verified by the manufacturer, and the compound structures are shown in Figure 10. AC is a molecular compatibility of four compounds at a certain ratio (patchouli alcohol: carvacrol: p-cymene: eucalyptol at 1:1:2:4, *w*/*w*), and this molecular compatibility only represents the concomitant application of the four compounds and does not involve a chemical reaction. Oseltamivir (the positive control drug), TPCK-treated trypsin, and dimethyl sulfoxide (DMSO) were obtained from Sigma Aldrich (St. Louis, MO, USA). Fetal bovine serum (FBS), Dulbecco’s Modified Eagle’s Medium (DMEM), penicillin–streptomycin solution, and phosphate buffer solution (PBS, pH 7.4) were purchased from Gibco (Waltham, MA, USA). The cholesterol and TG assay kits used in this study were obtained from Applygen (Beijing, China). The TNF-α, IL-6, IFN-γ, MCP-1, and MIP-1α ELISA kits were purchased from Proteintech (Wuhan, China). An IP-10 kit was obtained from Abcam (Cambridge, MA, USA). NLRP3, caspase-1, ASC, and GAPDH antibodies were purchased from CST (Danvers, MA, USA).

### 4.2. Cells, Viruses, and Culture Conditions

The human lung carcinoma A549 cells and canine kidney epithelial MDCK cells were purchased from the National Collection of Authenticated Cell Cultures (Shanghai, China). The A549 or MDCK cells were cultured in DMEM supplemented with 10% FBS, 100 U/mL penicillin, and 100 μg/mL streptomycin. The cells were then placed in a cell culture incubator at 37 °C with 5% CO_2_. The influenza A virus A/PR/8/34 (H1N1) strain was obtained from the Academy of Military Medical Sciences and propagated in MDCK cells supplemented with 1 μg/mL of N-tosyl-L-phenylalanine chloromethyl ketone (TPCK)-treated trypsin, as previously described [48].

### 4.3. Determination of the 50% Cytotoxicity Concentration (CC50)

The cytotoxicity of each group of drugs was evaluated using a Cell Counting Kit-81 (CCK-8, Beyotime, Shanghai, China) assay. The original medium was discarded, and 100 μL of DMEM containing 10% CCK-8 reagent was added to each well. The cell culture plate was then placed in a 37 °C, 5% CO_2_ cell culture incubator for 40 min. Next, the absorbance of each well was measured at 450 nm using a microplate reader, and the CC50 was calculated.

### 4.4. Anti-Influenza Virus Activity In Vitro

100 TCID50 of the influenza virus PR8 strain was utilized to infect the cells for 1 h, after which step the virus-containing medium was aspirated and replaced with the drug-containing medium, with a survival rate of at least 75%. The absorption values for each well were quantified after 48 h of incubation with the CCK-8 reagent, and the inhibitory concentration (IC50) and therapeutic index (TI) were calculated.

### 4.5. Animal Modeling and Treatment

BALB/c mice (male, 6–8 weeks old, 16–18 g) were purchased from SSPF Biotechnology Co., Ltd. (Beijing, China) [SCXK: 2019-0010]. All of the study protocols were conducted in accordance with the ethical guidelines and approved by the Ethics Committee for Animal Experiments of Beijing University of Chinese Medicine (Approval No. BUCM-4-2022062801-2094; Date: 1 April 2022). The mouse model of influenza virus infection has been described in the literature [3,4]. Briefly, the BALB/c mice were anesthetized and infected with 50 μL of the influenza virus PR8 strain (10^3^ TCID50) via a nasal drip, and the control group was administered an equal dose of PBS via inhalation. Two hours after viral infection, the control and model groups were given PBS, oseltamivir (25 mg/kg), the low-dose treatment AC (40 mg/kg), or the high-dose AC treatment (80 mg/kg) via gavage for 4 d (n = 10 mice per group). All of the mice were then euthanized, and tissue samples were collected for analysis on the 5th day after the virus challenge.

### 4.6. The Survival Study

After one week of acclimation, the BALB/c mice were immediately infected intranasally with the influenza virus strain PR8 (10^3^ TCID50). The mice were observed daily for 14 days after the challenge to determine their survival rate and to study the protective effect of the different treatments (with 10 mice per group) on the mice.

### 4.7. Histopathological Analysis

Segments of the right lung and colon were fixed in 4% formaldehyde solution and subsequently embedded into paraffin. Subsequently, the embedded specimens were sectioned into 4 μm thick sections, deparaffinized with xylene, and rehydrated with an ethanol gradient. Hematoxylin and eosin (HE) staining was performed on the tissue sections, which were then visualized on a NanoZoomerS60 slide scanner (Hamamatsu, Japan) using CaseViewer 2.4.0 software. HE staining of the tissue sections was analyzed using a double-blind method, with three section samples selected from each group.

### 4.8. The Enzyme-Linked Immunosorbent Assay (ELISA)

The levels of cytokines in the mouse lung and colon tissues were quantified using ELISA kits. In brief, the mouse lung and colon tissues were processed into tissue homogenates, and the resulting supernatants were obtained according to the manufacturer’s instructions. The expression levels of TNF-α, IL-6, IFN-γ, IP-10, MCP-1, and CCL3 were then analyzed.

### 4.9. Quantitative Real-Time PCR (RT-qPCR)

Total RNA was extracted from the mouse lung tissues using TRIzol (Invitrogen, Carlsbad, CA, USA) in accordance with the manufacturer’s instructions. RNA was reverse-transcribed into cDNA using the PrimeScript RT Reagent Kit (Takara, Shiga, Japan), and RT-qPCR was performed using the corresponding primers for the detection of the IAV NP or the target gene (ApoE). The relative mRNA expression levels were determined using the 2^−ΔΔCT^ method. The primer sequences utilized are presented in Appendix A.

### 4.10. Cholesterol and Triglyceride Enzymatic Assays

The levels of cholesterol and TGs in the cells and mouse lung tissues were quantified using a histocytometric enzyme assay kit (Applygen, Beijing, China). In accordance with the manufacturer’s instructions, the cells and mouse lung tissue were lysed. Following the incubation of the samples, the cholesterol and triglyceride (TG) contents in each group of samples were quantified based on the absorbance at 570 nm.

### 4.11. Flow Cytometry

Flow cytometry was used to determine the percentages of the CD4^+^ T-cell and CD8^+^ T-cell subsets in the mouse plasma and spleen tissues, respectively. In brief, peripheral blood lymphocytes and spleen cells were obtained separately, and surface-labeled antibodies (CD3-BV510, CD4-FITC, and CD8-APC, 1:100) were added. The labeling reaction was performed in the dark. Subsequently, the cell suspensions obtained through washing and centrifugation were subjected to a second examination for T-cell sorting.

### 4.12. The Immunofluorescence Assay

In this study, we determined the expression of F4/80, iNOS, CD206, and ROS in the lung tissues and the expression of ZO-1 and occludin in the colon tissues from the mice in each group. The quantity of LDs in the IAV-infected mice and cells was determined separately. These procedures are described in the Appendix A.

### 4.13. Western Blotting

Total proteins were extracted from the mouse lung tissue samples using a RIPA lysate (Beyotime, Shanghai, China) containing 1% PMSF and 1% protease/phosphatase inhibitor, and the protein concentrations were quantified using a BCA protein assay kit (Beyotime, Shanghai, China). Subsequently, equal amounts of the protein samples were separated on a 10% SDS-PAGE gel, and the protein blot was transferred onto a 0.45 μm PVDF membrane (Millipore, Burlington, MA, USA). The PVDF membrane was closed using 5% BSA-TBST for one hour at room temperature and incubated with the primary antibody (dilution ratio 1:1000) at 4 °C overnight. The primary antibodies utilized included GAPDH, NLRP3, Caspase1, and ASC. The secondary antibody, a goat anti-rabbit IgG (H+L) HRP conjugate (Jackson, Lansing, MI, USA), was diluted using 5% BSA-TBST (dilution ratio = 1:10,000) and incubated at room temperature for 1h. The membranes were then exposed to ECL reagent (Absin, Shanghai, China) to visualize the protein blot. The blots were analyzed using the ImageJ software (1.8.0), with GAPDH serving as the internal control.

### 4.14. Gut Microbiota Analysis

We performed DNA extraction and detection on each group of mouse samples, followed by PCR amplification and purification and upsequencing and a data analysis after the quality was confirmed. After machine sequencing, the bipartite data were spliced using an overlap, and quality control and chimera filtering were performed to obtain high-quality clean data, and then the final ASV feature list, as well as feature sequences, were obtained after denoising using Divisive Amplicon Denoising Algorithm (DADA2), and then a further diversity analysis, species taxonomic annotation and a differential analysis, and other data mining steps were performed.

### 4.15. Lipid Metabolomics Analysis

The serum samples from each group of mice were extracted using organic reagents and then subjected to mass spectrometry scanning in the positive and negative ion modes. XCMS software (4.0) was used for the peak extraction and quality control, while metaX software (4.0) was used for metabolite identification. The identified metabolites were subsequently subjected to a common analysis. The metabolites were then subjected to a quantitative analysis, a sample correlation analysis, and a difference analysis, and a series of functional analyses of differential metabolites, such as a KEGG functional enrichment analysis, an interaction network analysis, and a metabolite correlation analysis, were performed for the differential metabolites.

### 4.16. Statistical Analysis

Statistical analyses were conducted using GraphPad Prism software (version 8.2), and a one-way or two-way analysis of variance (ANOVA) was used to assess significant differences between groups. If the data did not satisfy the normality assumptions of the ANOVA, a further nonparametric statistical analysis was performed using the Kruskal-Wallis test. Intragroup correlations between differentially abundant gut flora and metabolites were evaluated using Spearman’s correlation coefficient, and all of the data are expressed as the mean ± standard deviation (SD). A *p* value < 0.05 was considered to indicate statistical significance.

## 5. Conclusions

In this study, we demonstrated that AC inhibits influenza virus in the lung tissues of model mice and improves their survival. Meanwhile, it was found that AC could ameliorate the inflammatory injury in the lung and colon tissues of the model mice through the lung–intestinal axis and regulate the excessive immune response, calm the cytokine storm, and effectively regulate intestinal flora disorders in the model mice. Subsequently, AC also effectively regulated the IAV-induced lipid metabolism disorders in the mice with ALIs by directly suppressing their cholesterol and TG levels, thereby reducing the abnormal accumulation of LDs. However, this study has some limitations, as potential direct interactions between AC and host or viral proteins were not clarified in this study, and the effects on inflammation and lipid metabolism regulation were only verified in the model of IAV-induced ALI and will need to be verified further in other animal models, such as an ALI model of SARS-CoV-2 infection or lipopolysaccharide-induced ALI. In the future, the precise mechanism through which AC regulates the metabolism of LDs and the nodal molecules through which AC inhibits inflammatory signaling and remodels lipid metabolism in the treatment of ALIs will need to be explored further. In conclusion, AC could ameliorate IAV-induced ALIs through multiple pathways, such as antiviral, anti-inflammatory, and LD regulation pathways, which provides a promising strategy for the clinical application of aromatic Chinese medicine in the treatment of ALIs (Figure 10).

## Figures and Tables

**Figure 1 pharmaceuticals-18-00468-f001:**
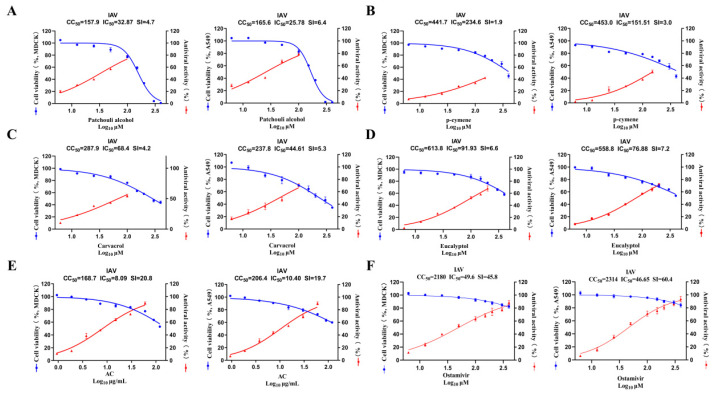
**In vitro safety and antiviral activity.** (**A**–**F**) Dose–response curves for half-maximal inhibitory concentration (IC50), half-maximal cytotoxic concentration (CC50), and selectivity index (SI) of patchouli alcohol, carvacrol, p-cymene, eucalyptol, AC, and oseltamivir against IAV. The cytotoxicity of six drug groups against the MDCK and A549 cells was determined using a CCK8 assay (n = 3).

**Figure 2 pharmaceuticals-18-00468-f002:**
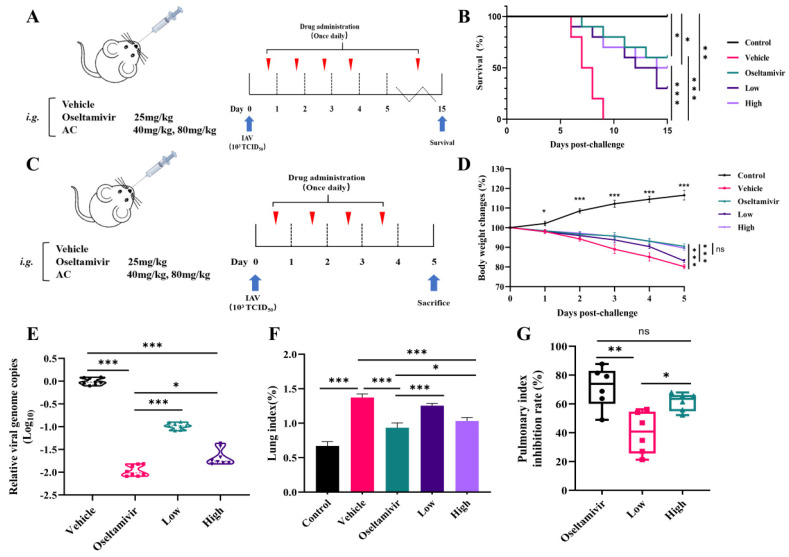
**AC protects mice infected with influenza virus.** (**A**) Scheme of once-daily dosing for 14 consecutive days after infecting mice with the 10^3^TCID50 IAV PR8 strain via nose drops (n = 10). (**B**) Survival rates (expressed as percentages) of IAV-challenged mice through treatment with oseltamivir and AC (n = 10). (**C**) Scheme of once-daily dosing for 4 consecutive days after using nose drops infected with the 10^3^TCID50 IAV PR8 strain in mice (n = 10). (**D**) Effect of treatment with oseltamivir and AC on the recovery of body weight in IAV-infected mice (n = 10). (**E**) Reduction in IAV genome copy number in lungs sampled from IAV-challenged mice through treatment with AC (n = 8). (**F**,**G**) Lung index and lung index inhibition in H1N1 mice after AC treatment (n = 6). The data are presented as the mean ± SD, * *p* < 0.05, ** *p* < 0.01, *** *p* < 0.001, ns: not significant.

**Figure 3 pharmaceuticals-18-00468-f003:**
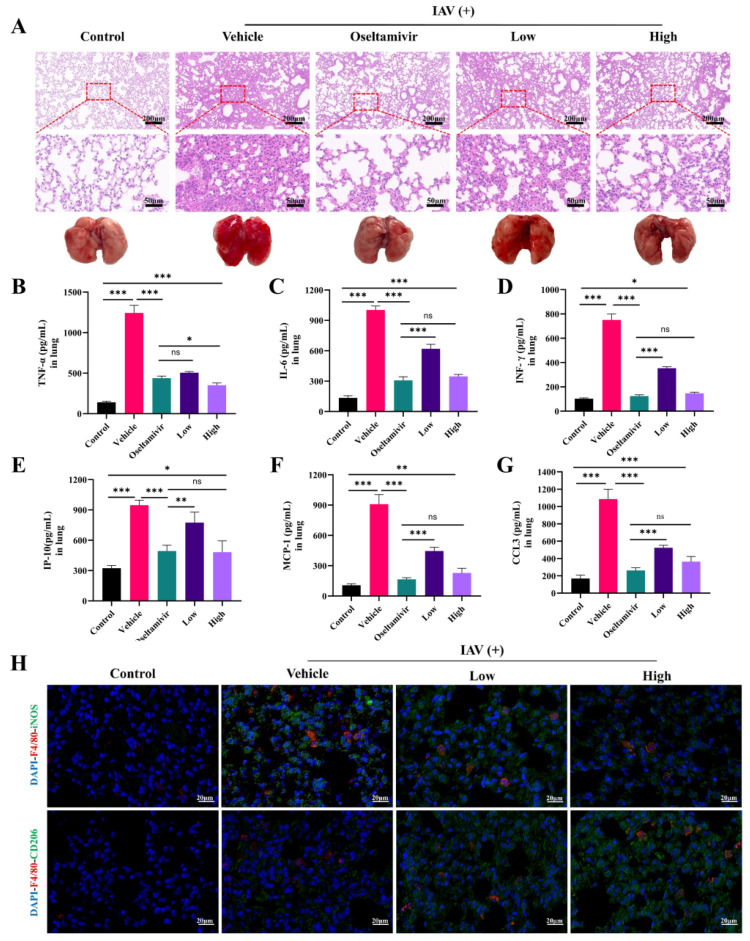
**AC attenuates IAV-induced pulmonary inflammation in mice with ALIs.** (**A**) Representative images showing histopathological changes in the lung tissues and their amelioration following the AC treatment (upper panel scale bar = 200 μm, lower panel scale bar = 50 μm, n = 3). (**B**–**G**) The expression of TNF-α, IL-6, INF-γ, IP-6, MCP-1, and CCL3 in the pulmonary tissue was evaluated in each group (n = 6). (**H**) Lung tissue F4/80 (red) and iNOS (green) expression and lung tissue F4/80 (red) and CD206 (green) expression after AC treatment of the IAV-infected mice (scale bar = 20 μm, n = 3). The data are presented as the mean ± SD, * *p* < 0.05, ** *p* < 0.01, *** *p* < 0.001, ns: not significant.

**Figure 4 pharmaceuticals-18-00468-f004:**
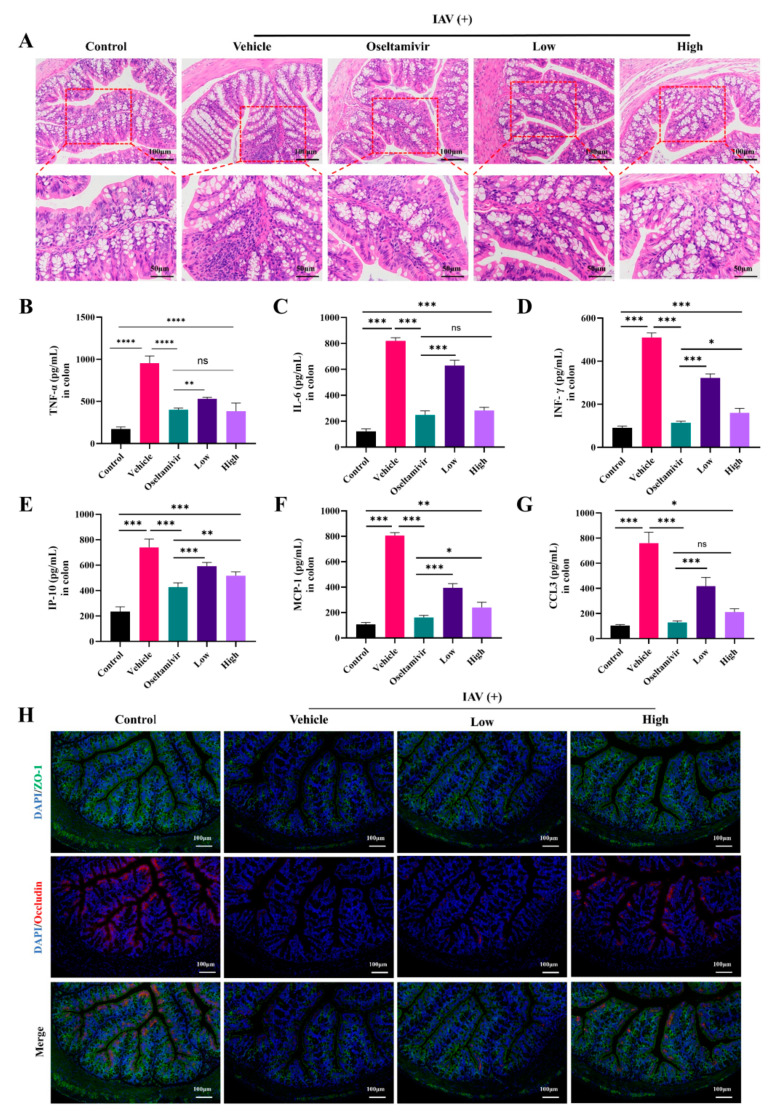
**AC attenuates IAV-induced colonic lesions in mice with ALIs.** (**A**) Representative images of colonic histologic lesions. Improvement in colonic histologic lesions through AC therapy (upper panel scale bar = 100 μm, lower panel scale bar = 50 μm, n = 3). (**B**–**G**) The expression of TNF-α, IL-6, INF-γ, IP-6, MCP-1, and CCL3 in the colon tissue was evaluated in each group (n = 6). (**H**) Expression of ZO-1 (green) and occludin (red) in colonic tissues of mice with IAV-induced ALIs after AC treatment (scale bar = 100 μm, n = 3). The data are presented as the mean ± SD, * *p* < 0.05, ** *p* < 0.01, *** *p* < 0.001, **** *p* < 0.0001, ns: not significant.

**Figure 5 pharmaceuticals-18-00468-f005:**
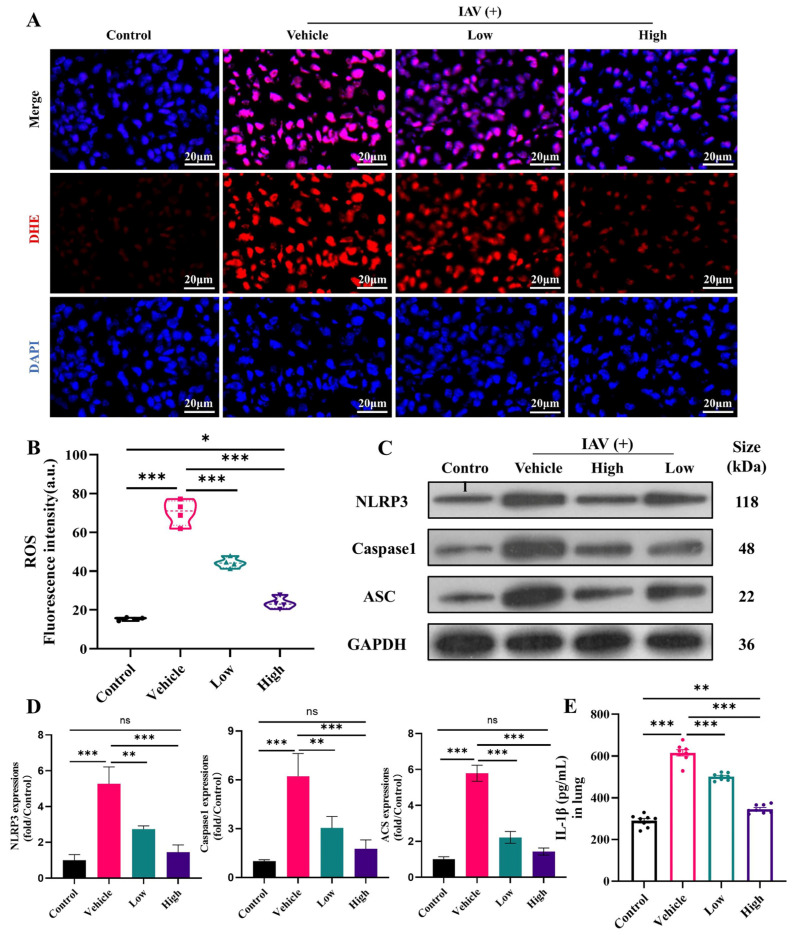
**AC inhibits ROS and the NLRP3 inflammasome in mice with IAV-induced ALIs.** (**A**,**B**) AC inhibits ROS expression (scale bar = 20 μm) and fluorescence quantification (n = 4) in the lung tissues of mice with IAV-induced ALIs. (**C**,**D**) Lung tissue WB bands and relative protein expression of NLRP3/ASC/caspase-1-related proteins (n = 3). (**E**) IL-1β expressions in the lungs (n = 8). The data are presented as the mean ± SD, * *p* < 0.05, ** *p* < 0.01, *** *p* < 0.001, ns: not significant.

**Figure 6 pharmaceuticals-18-00468-f006:**
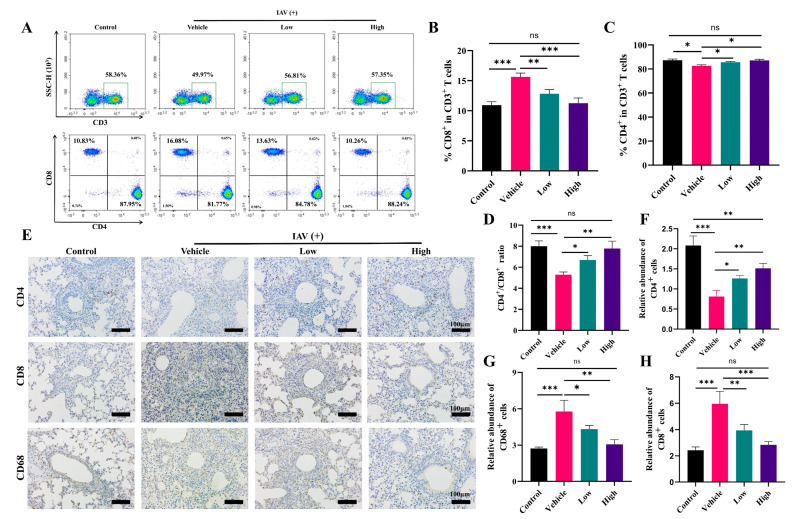
**AC suppresses immune cell infiltration in mice with IAV-induced ALIs.** (**A**) Effect of AC on T-lymphocytes in peripheral blood of mice with IAV-induced ALIs (The green box represents the proportion of CD3^+^ T cells, n = 3). (**B**–**D**) The CD8+ T cells, CD4+ T cells, and CD4+/CD8+ ratio in the peripheral blood undergo alterations following the AC treatment (n = 3). (**E**) The regulation of AC on the expression of CD4+, CD8+, and CD68+ in the lung tissues of mice with IAV-induced ALIs (scale bar = 100 μm, n = 3). (**F**–**H**) The expression levels of CD8+, CD4+, and CD68+ in the lung tissue following the AC treatment were evaluated (n = 3). The data are presented as the mean ± SD, * *p* < 0.05, ** *p* < 0.01, *** *p* < 0.001, ns: not significant.

**Figure 7 pharmaceuticals-18-00468-f007:**
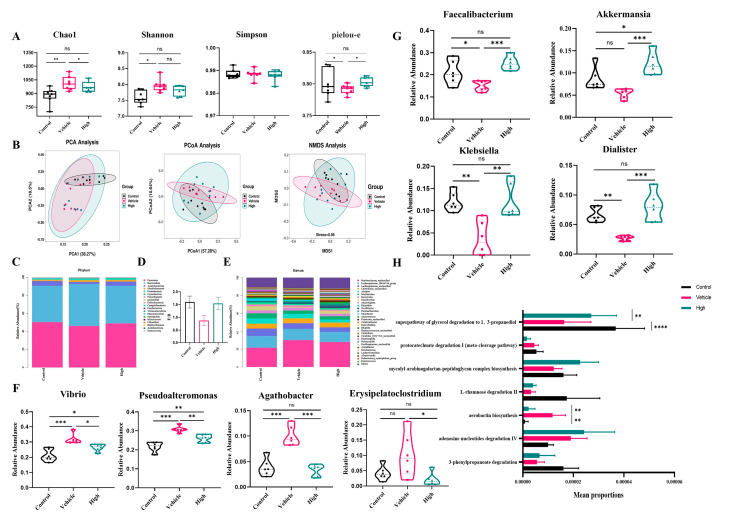
**AC’s modulation of the gut microbiota in mice with IAV-induced ALIs and analysis of species differences.** (**A**) α-diversity analysis (n = 6) and (**B**) PCA, PCoA of the weighted unifrac, and NMDS analysis based on the Bray–Curtis distance algorithm for the control, vehicle, and high-dose AC treatment groups (n = 9). (**C**) Differences in the gut microbiota at the phylum level. (**D**) The ratio of *Firmicutes/Bacteroidetes* (F/B) in the three groups of mice (n = 6). (**E**) The relative abundance of the differential microbiomes between the control and vehicle groups at the genus level was found to be significantly regulated by AC. (**F**) Inhibitory effect of AC on conditionally pathogenic intestinal bacteria after IAV infection (n = 6). (**G**) Effect of AC on intestinal probiotics after IAV infection (n = 6). (**H**) Functional enrichment of differential flora at the pathway level. The data are presented as the mean ± SD, * *p* < 0.05, ** *p* < 0.01, *** *p* < 0.001, **** *p* < 0.0001, ns: not significant.

**Figure 8 pharmaceuticals-18-00468-f008:**
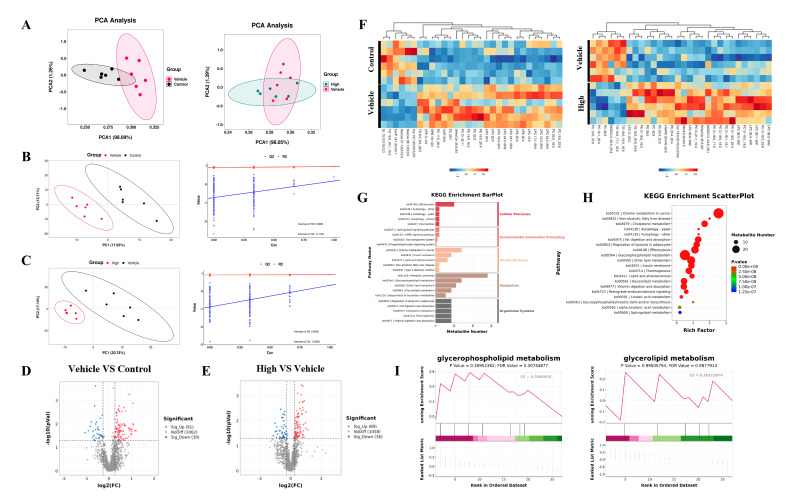
**Lipid metabolism analysis of the effect of AC on sera from mice with IAV-induced ALIs.** (**A**) PCA and PLS-DA (**B**,**C**) score plots of the serum lipid metabolite profiles between the different groups. Volcano plots of differential metabolites between (**D**) control and vehicle groups and (**E**) high-dose and vehicle groups, where red represents upregulated significantly differentially expressed metabolites, blue represents downregulated significantly differentially expressed metabolites, and gray dots represent non-significantly differentially expressed metabolites. (**F**) Heat map of differential metabolites between the control and vehicle and AC (high-dose) and vehicle groups. (**G**) KEGG enrichment analysis, (**H**) enrichment scatter plot, and KEGG-enriched ES fold plot (**I**) showed the changes in the lipid metabolic process in the serum after the AC treatment (Red represents genes enriched in the AC treatment group, and green represents genes enriched in the model group). The data are presented as the mean ± SD, n = 6.

**Figure 9 pharmaceuticals-18-00468-f009:**
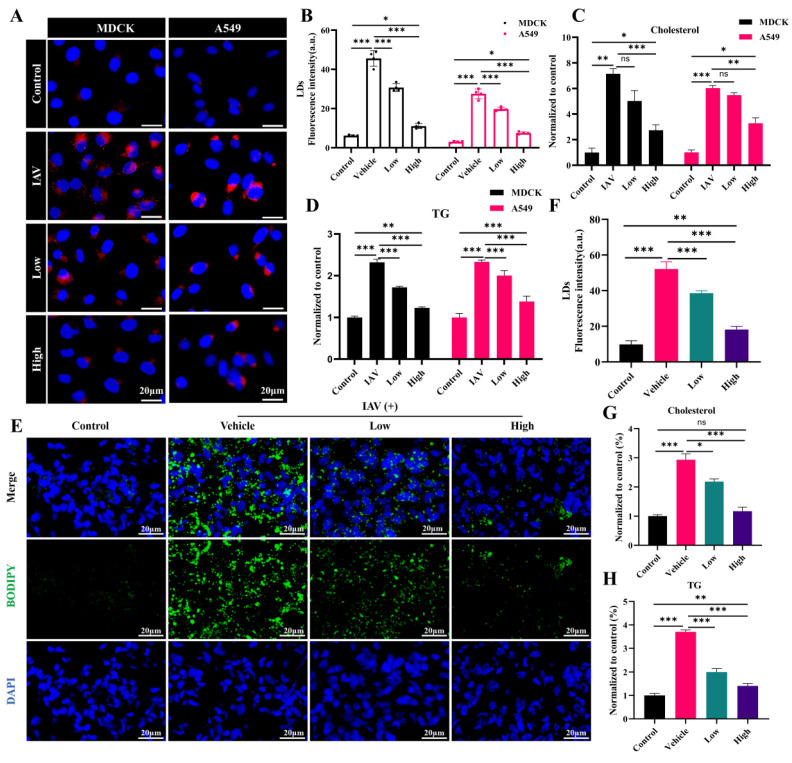
**AC suppressed the abnormal IAV-induced accumulation of LDs.** (**A**,**B**) The effect of AC on the accumulation of LDs. (**C**) Cholesterol and (**D**) TG levels in the MDCK and A549 cells after IAV infection. (**E**,**F**) The effect of AC on the accumulation of LDs and (**G**) cholesterol and (**H**) TG levels in the lung tissue of mice with IAV-induced ALIs. The data are presented as the mean ± SD, n = 4. * *p* < 0.05, ** *p* < 0.01, *** *p* < 0.001, ns: not significant.

**Figure 10 pharmaceuticals-18-00468-f010:**
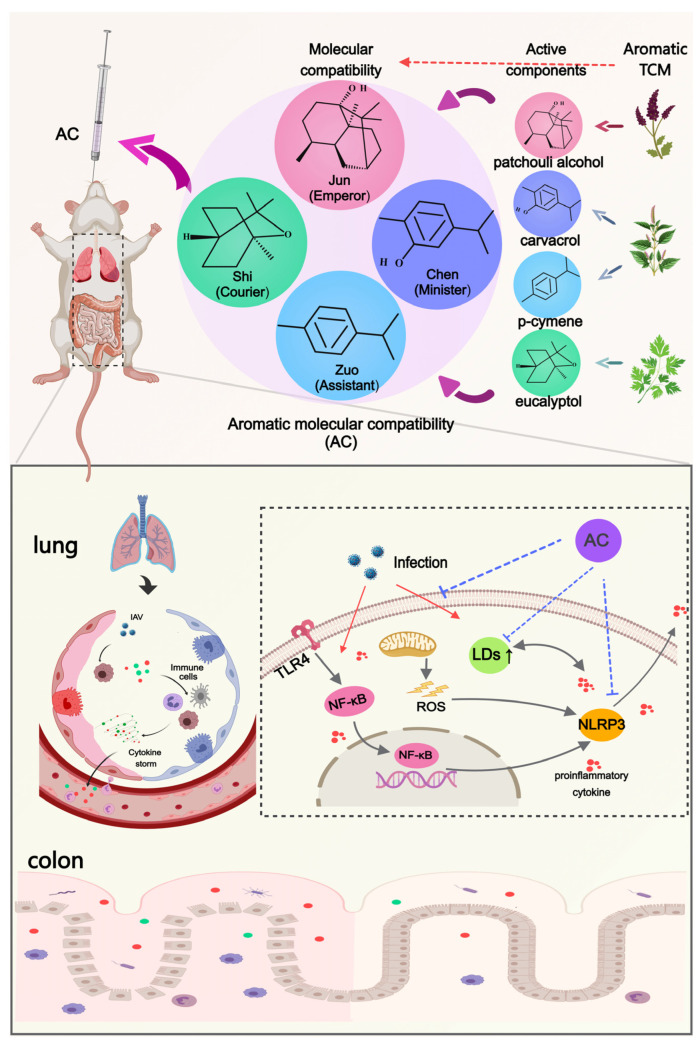
Aromatic molecular compatibility (AC) obtained using four known active compounds obtained from aromatic Chinese medicines by applying the theory of Chinese medicine and molecular compatibility. The protective effects of AC on mice with IAV-induced ALIs were illustrated based on multiple pathways, including antiviral and anti-inflammatory pathways and the modulation of the lung–intestinal axis and lipid droplet metabolism.

## Data Availability

The original contributions presented in this study are included in the article/Appendix A. Further inquiries can be directed to the corresponding authors.

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
