# Peer review of "Aromatic Molecular Compatibility Attenuates Influenza Virus-Induced Acute Lung Injury via the Lung–Gut Axis and Lipid Droplet Modulation"

_pharmaceuticals, 2025, doi:10.3390/ph18040468_

Round 1
Reviewer 1 Report
Comments and Suggestions for Authors
In folk medicine, essential oils and aromatic molecules are currently used as therapeutic agents. On the other hand, numerous studies show that these natural compounds exert significant biological and pharmacological effects on different experimental models, mainly due to their lipophilic nature. Therefore, the characterization and knowledge of the mechanism of action of the various aromatic compounds allows a more targeted application in different fields of application.
In this study, the therapeutic effects and mechanisms of an aromatic molecular composition (AC) in mice with acute lung injury (ALI) induced by influenza A virus (IAV) were evaluated by using different biological techniques.
The topic covered in the manuscript is interesting and deserves to be explored in depth.
The Introduction section correctly reports the reasons that moved the presented research.
The selection of literature is appropriate throughout the manuscript.
The Materials and Methods section is written in a sufficiently clear manner.
Overall, the results and discussion are well presented. However, there are some points to be corrected and clarified, as reported below.
The title of a scientific article should be simple, clear, not emphatic. Please take this into account.
Line 50: Replace the word "current" with "recent".
Lines 98-109: This section should be eliminated and replaced with sentences that only report the purpose of the work.
Lines 115-116: Here and in other places in the manuscript the authors report the sentence “Error! Reference source not found…” What does it mean?
Please check that the Figures and Tables presented are also correctly referenced in the written text.
In the Conclusions section, the authors should report the limitations of the study and future investigations to be carried out.
Author Response
Comments 1: The title of a scientific article should be simple, clear, not emphatic. Please take this into account.
Response 1: Thank you very much for your positive suggestions for this study. In response to your suggestions, we have changed the title to “Aromatic Molecular Compatibility Attenuates Influenza Virus-Induced Acute Lung Injury via Lung-Gut Axis and Lipid Droplets Modulation”, which can be seen in lines 2-4 of the revised manuscript, marked in red.
Comments 2: Line 50: Replace the word "current" with "recent".
Response 2: We sincerely appreciate your suggestions for this manuscript. According to your comment, we have changed “current” to “recent”, which can be found on line 51 and is highlighted in red.
Comments 3: Lines 98-109: This section should be eliminated and replaced with sentences that only report the purpose of the work.
Response 3: Thank you for your valuable suggestions on our research. Based on your comments, we have reworked this section and corrected it to “In this study, we employed a variety of biological techniques to validate the therapeutic effects and mechanisms of AC on IAV-induced ALI mice. First, the inhibitory effect of AC on influenza virus was evaluated by in vitro in vivo experiments. Subsequently, based on the lung-gut axis theory, we investigated the effects of AC on inflammatory injury in lung and colon tissues and gut flora in model mice. Meanwhile, the effect of AC on lipid metabolic remodeling was investigated by metabolomics and verified by lipid droplets”, which can be found in lines101-107 of the revised draft and is highlighted in red.
Comments 4: Lines 115-116: Here and in other places in the manuscript the authors report the sentence “Error! Reference source not found…” What does it mean?
Response 4: We apologize for failing to use the correct reference formatting in the previously submitted manuscript. In this revised manuscript (Word version), we have checked that all references are formatted in accordance with the requirements of the journal. This can be seen in line 732 of the revised manuscript under “References”.
Comments 5: Please check that the Figures and Tables presented are also correctly referenced in the written text.
Response 5: Thank you very much for your advice. According to your comments, we checked the citation of Figures and Tables in the manuscript, and Figures 1-Figure 10 in the article are correctly cited. Among them, Figures S1-S4 and Tables S1-S2 are in the supplementary material. The citations in the Figures and Tables are bolded in black in the revised manuscript.
Comments 6: In the Conclusions section, the authors should report the limitations of the study and future investigations to be carried out.
Response 6: Thank you for your valuable suggestions on our research. Following your comments, we have added the limitations of this study and elements for future investigations to be conducted “However, this study has some limitations, as the potential direct interactions between AC and host or viral proteins were not clarified in this study, and the effects on inflammation and lipid metabolism regulation were only verified in the IAV-induced ALI model, and will need to be further verified in other animal models, such as the ALI model of SARS-CoV-2 infection or lipopolysaccharide-induced ALI. In the future, the precise mechanism by which AC regulates the metabolism of LDs and the nodal molecules by which AC inhibits inflammatory signaling and remodels lipid metabolism in the treatment of ALI will need to be further explored” in lines 691-698 of the revised manuscript and in red font.
Reviewer 2 Report
Comments and Suggestions for Authors
I have reviewed the manuscript by Li et al. Authors have clear aims, sound methodology and major findings. However, before manuscript can be considered, there are some points which needs to be addressed and rectify.
- Firstly, “Aromatic Molecular Compatibility” is a new term. For any reader it will take time to understand what basically the paper is about, and what this term is. I think you have to simplify the title or try to use some appropriate terminology in the brackets for easy understanding.
- In the abstract you are suggesting that AC treatment directly regulates intestinal flora and LD accumulation, while in the manuscript you are describing an indirect mechanism via modulation of cholesterol and triglyceride metabolism. You have to clarify this and make it consistent in the manuscript on whether the effect is direct or indirect via gut flora.
- What is the rationale behind choosing MDCK and A549 cells for the cytotoxicity assay. It is understandable that MDCK cells are standard for influenza virus propagation, but what is the point and relevance of choosing A549 cells for antiviral assessment? You have to explain this meaningfully.
- In TCID50 assay, you have mentioned about observing cells on daily basis. Please provide the exact duration for observation until CPE cessation to enhance reproducibility.
- Similarly, you have to specify clearly the criterion for defining "cytopathic lesion cessation".
- Any specific reason that why the dosing was conducted for 4 or 14 days? Is it based on previous literature literature?
- You have to mention number of replicates, statistical tests applied in Table S2 (Supplementary material).
- The antiviral efficacy of AC relative to oseltamivir is clearly presented. However, the difference in SI values could be emphasized and further discussed in terms of practical therapeutic significance.
- I can see that AC reduced viral load, yet acknowledges a lower antiviral effect compared to oseltamivir. Please discuss why AC’s antiviral effect is lower yet comparable in terms of survival and inflammatory protection, suggesting non-antiviral therapeutic mechanisms.
- In histopathological analysis, how many fields analyzed, blinded or not?
- Figure S5 needs more explicit labeling—please clearly mark which bands correspond to which proteins and specify molecular weights clearly for transparency. You have provided original blots for reference; here you can clearly label the bands. Also, the legend needs to be more explanatory. It is difficult to understand why 3 images for each? Are they replicates (n=3)?
- Why certain bacterial genera were selected for specific analysis? Are these relevant to influenza-induced inflammation? I do not understand here the objective. You have to clarify this.
- Explain if lipid metabolism changes observed (particularly cholesterol and TG levels) directly correlate with microbiota changes, as implied by correlation analysis, or whether there are other potential intermediate mechanisms? You can include this in discussion.
- You have mentioned your few limitations in the discussion section. It’s better either to add in conclusion or have separate heading “Limitations of the study”. You have not addressed the potential direct interactions between aromatic molecules and host or viral proteins.
- You have to address data presentation errors such as repeated "Error! Reference source not found" references. Most of the places this is appearing. You have to correct it.
- No standardisation of the figure legends, without clear experimental conditions and number of replicates. Poor and non-explanatory figure legends.
- "The therapeutic effect of AC was verified by evaluating the antiviral effect of AC on model mice, and the improvement of lung and colon inflammatory injury, oxidative stress and NLRP3 inflammasome."
It’s better to revise the sentence to:
"The therapeutic effect of AC was verified by evaluating antiviral efficacy in mouse models, including improvements in lung and colon inflammation, oxidative stress, and suppression of the NLRP3 inflammasome."
- Significant structural and grammar issue. Please rectify. Some example below:
- "Our team screened four volatile compounds from aromatic Chinese medicines and defined the concomitant application of the four compounds as aromatic molecular compatibility (AC) through molecular compatibility theory, but the therapeutic effect and mechanism of AC were unclear."
It’s better to revise the sentence to:
"Our team previously identified four volatile compounds from aromatic Chinese medicines. Based on molecular compatibility theory, we defined their combination as aromatic molecular compatibility (AC), though its therapeutic effects and underlying mechanisms remain unclear."
2. "Given the significant morbidity and mortality associated with influenza pandemics each year and the potential for influenza to have a similar impact as that of COVID-19..."
Add comma after "each year”
3. The phrase "Error! Reference source not found." frequently appears, indicating citation or referencing issues throughout the manuscript.
4. "This study employed a mouse model of influenza virus infection induced by administration of nasal drops of the IAV……..
It’s better to revise the sentence to:
"This study employed a mouse model infected intranasally with Influenza A virus (IAV) PR8 strain (10³ TCIDâ‚…â‚€, 50 μL)."
5. "mortality commenced on the sixth day of treatment in the vehicle group compared with the control group and persisted until the ninth day."
It’s better to revise the sentence to:
"Mortality began on day 6 in the vehicle-treated group compared with controls and continued until day 9 post-infection."
6. Fig. 3 legend: "Representative images of histologic lesions in the lungs. Amelioration of histologic lung lesions by treatment with AC."
It’s better to revise the sentence to:
"Representative images showing histopathological changes in lung tissues and their amelioration following AC treatment."
- There are many language issues in the manuscript. Authors have to take professional English editing service.
7. In section “Animal Modeling and Treatment”: "The mouse model of influenza virus infection has been described in the literature."
Reference is needed here
8. Verify all abbreviations used in figures/tables, and are clearly explained within figure legends or table notes. Majority of the abbreviations used in figures and tables are not mentioned in the abbreviation list.
9. There is a noticeable repetition of citations referencing previous work using "Error! Reference source not found," making it unclear exactly which references correspond to the authors' previous work. This error you have to correct to confirm if you are mentioning any othere reference beyond ref no. 41.
Comments on the Quality of English LanguageThere are significant level of language issues (grammatical, punctuations, syntax, unnecessarily compliocated sentences etc.) in the manuscript. Authors have to take professional English editing service.
Author Response
Comments 1: Firstly, “Aromatic Molecular Compatibility” is a new term. For any reader it will take time to understand what basically the paper is about, and what this term is. I think you have to simplify the title or try to use some appropriate terminology in the brackets for easy understanding.
Response 1: Thank you for your valuable suggestions on our research. As you mentioned, “Aromatic Molecular Compatibility” is a new term. It is the result of our team's screening of four volatile compounds from aromatic Chinese medicines, and defining the combination application of the four compounds as Aromatic Molecular Compatibility (AC) through TCM compatibility theory and molecular compatibility theory. We have explained “Aromatic Molecular Compatibility” in lines 95-98 of the revised manuscript, and the Title (lines 2-4) of the article has also been revised and highlighted in red.
Comments 2: In the abstract you are suggesting that AC treatment directly regulates intestinal flora and LD accumulation, while in the manuscript you are describing an indirect mechanism via modulation of cholesterol and triglyceride metabolism. You have to clarify this and make it consistent in the manuscript on whether the effect is direct or indirect via gut flora.
Response 2: Thank you for your valuable suggestions on our research. We apologize for failing to clearly elucidate the effects of AC's on gut flora and lipid metabolism.We clarified that AC effectively modulated intestinal flora disorders and lipid metabolism in model mice, which could reduce the abnormal accumulation of LDs by directly decreasing cholesterol and triglyceride levels after influenza virus infection.Also, in order to be consistent with the results in the abstract, we have reworded lines 379-383 and 688-691 in the revised manuscript and marked them in red font, but these changes do not affect the conclusions of this study.
Comments 3: What is the rationale behind choosing MDCK and A549 cells for the cytotoxicity assay. It is understandable that MDCK cells are standard for influenza virus propagation, but what is the point and relevance of choosing A549 cells for antiviral assessment? You have to explain this meaningfully.
Response 3: Thank you for your review and valuable comments on this study design. Regarding the scientific rationale for the selection of the MDCK and A549 cell lines and the significance of their complementarity, we provide here the following detailed description.
There is no question that MDCK cells are a classic model for influenza virus research, but the selection of A549 was based on the following key considerations: (1) A549 cells are derived from human alveolar basal epithelium, which retains a variety of human lung cell properties and expresses a specific sialic acid receptor that can be infected by PR8 virus; (2) MDCK is a canine-derived cell, and its drug metabolizing enzymes, membrane transport proteins, and signaling pathways differ from those of human cells. A549 can verify the activity of AC in human cells, avoiding false-positive/false-negative results due to cross-species differences.
Also, several studies [1-3] have reported the use of A549 cells in antiviral experiments. Therefore, the purpose of using MDCK and A549 cells was to realize the direct inhibitory effect of AC on virus in MDCK cells and to further validate the antiviral efficacy of AC in human-derived cells in A549 cells. We suggest that the combined use of MDCK and A549 cells is not redundant, but rather reveals the antiviral potential and safety of AC in a more comprehensive manner and provides a reliable basis for subsequent studies.
1. Luo, J.; Gao, Y.; Guo, W.; Zhao, S.; Li, L.; Zhang, Z.; Gao, R. Spleen tyrosine kinase inhibitor R406 has both antiviral and anti-inflammatory effects on severe influenza A infection. Journal of medical virology 2024, 96, e29678, doi:10.1002/jmv.29678.
2. Bai, L.; Zhao, Y.; Dong, J.; Liang, S.; Guo, M.; Liu, X.; Wang, X.; Huang, Z.; Sun, X.; Zhang, Z.; et al. Coinfection with influenza A virus enhances SARS-CoV-2 infectivity. Cell research 2021, 31, 395-403, doi:10.1038/s41422-021-00473-1.
3. Wu, Z.; Zhao, C.; Ai, H.; Wang, Z.; Chen, M.; Lyu, Y.; Tong, Q.; Liu, L.; Sun, H.; Pu, J.; et al. A Susceptible Cell-Selective Delivery (SCSD) of mRNA-Encoded Cas13d Against Influenza Infection. Advanced science (Weinheim, Baden-Wurttemberg, Germany) 2025, 12, e2414651, doi:10.1002/advs.202414651.
Comments 4: In TCID50 assay, you have mentioned about observing cells on daily basis. Please provide the exact duration for observation until CPE cessation to enhance reproducibility.
Response 4: Thank you for your careful review of this study and your comments have improved the seriousness of this study. The growth of the cells was observed and recorded every day, the cytopathic effect (CPE) was observed every 12 hours, and the TCID50 calculation was performed after the cessation of cytopathic lesions (72 hours in this study). This study was conducted in strict accordance with the Standard Operating Procedures for Chinese National Influenza Center. We have also described the experiment on lines 34-40 in the Supporting Material and highlighted them in red.
Comments 5: Similarly, you have to specify clearly the criterion for defining "cytopathic lesion cessation".
Response 5: Thank you for your careful review of this study and your comments have improved the seriousness of this study. Cytopathic lesion cessation was defined as no further change in the phenomenon of shedding or reduction of virus-infected cells for a period of time generally ranging from 72-96 hours, which in this study was 72 hours. We have clarified this in lines 36-38 of the Supporting Material and highlighted them in red.
Comments 6: Any specific reason that why the dosing was conducted for 4 or 14 days? Is it based on previous literature literature?
Response 6: Thank you for your valuable comments on this study, the reasons for the different days of animal modeling are as follows.
Regarding the continuous 14-day dosing experiment in mice with influenza pneumonia, we focused on verifying whether AC could improve the survival rate [3-5] of model mice.
The setting of 4 days of continuous administration in a mouse model of influenza virus pneumonia, was to assess the effect of AC on viral replication and acute inflammatory response [6-8] in model mice during the acute infection period [9].
4. Chakraborty, S.; Cheng, B.Y.; Edwards, D.L.; Gonzalez, J.C.; Chiu, D.K.; Zheng, H.; Scallan, C.; Guo, X.; Tan, G.S.; Coffey, G.P.; et al. Sialylated IgG induces the transcription factor REST in alveolar macrophages to protect against lung inflammation and severe influenza disease. Immunity 2025, 58, 182-196.e110, doi:10.1016/j.immuni.2024.10.002.
5. Ling, L.; Ren, A.; Lu, Y.; Zhang, Y.; Zhu, H.; Tu, P.; Li, H.; Chen, D. The synergistic effect and mechanisms of flavonoids and polysaccharides from Houttuynia cordata on H1N1-induced pneumonia in mice. Journal of ethnopharmacology 2023, 302, 115761, doi:10.1016/j.jep.2022.115761.
6. Zhang, Y.; Xu, W.F.; Yu, Y.; Zhang, Q.; Huang, L.; Hao, C.; Shao, C.L.; Wang, W. Inhibition of influenza A virus replication by a marine derived quinolone alkaloid targeting virus nucleoprotein. Journal of medical virology 2023, 95, e28499, doi:10.1002/jmv.28499.
7. Cao, S.; Gao, S.R.; Ni, C., Zi-HanXu, Ying-LiPang, BoChen, Meng -PingZhang, YuGuo, Shan-ShanShi, Yu-JingNi, Li-QiWang, KunZhao, Rong-HuaCui, Xiao-LanBao, Yan-Yan %J traditional medicine research. Pudilan Xiaoyan oral liquid regulates tissue inflammation and apoptosis in mice with influenza virus pneumonia. 2024, 9.
8. Lian, B.; He, S.; Jiang, H.; Guo, Y.; Cui, X.; Jiang, T.; Su, R.; Chen, Y.; Zhao, C.; Zhang, M.; et al. Qin-Qiao-Xiao-Du formula alleviate influenza virus infectious pneumonia through regulation gut microbiota and metabolomics. Frontiers in medicine 2022, 9, 1032127, doi:10.3389/fmed.2022.1032127.
9. Ichinohe, T.; Lee, H.K.; Ogura, Y.; Flavell, R.; Iwasaki, A. Inflammasome recognition of influenza virus is essential for adaptive immune responses. The Journal of experimental medicine 2009, 206, 79-87, doi:10.1084/jem.20081667.
Comments 7: You have to mention number of replicates, statistical tests applied in Table S2 (Supplementary material).
Response 7: Thank you for your valuable comments on this study. We have added the number (n=3) of experiments and statistical differences in Table S2 in the Supporting Materials (lines 176-179) and labeled them in red font .
Comments 8: The antiviral efficacy of AC relative to oseltamivir is clearly presented. However, the difference in SI values could be emphasized and further discussed in terms of practical therapeutic significance.
Response 8: Thank you for your valuable comments on this study. Based on your comments, we have provided additional explanations in lines 119-127 of the revised draft and marked them in red font.
Comments 9: I can see that AC reduced viral load, yet acknowledges a lower antiviral effect compared to oseltamivir. Please discuss why AC’s antiviral effect is lower yet comparable in terms of survival and inflammatory protection, suggesting non-antiviral therapeutic mechanisms.
Response 9: Thank you for taking your valuable time to read this study and for your meaningful comments. As you have seen in the study results, AC has a lower antiviral effect compared to oseltamivir, but its survival and anti-inflammatory effect effects are similar to those of oseltamivir. This is an interesting phenomenon, and that's why we conducted a follow-up study to investigate the mechanism of efficacy of AC. According to your comments we have discussed this phenomenon, adding to it in lines 463-467 and labeling them in red.
Comments 10: In histopathological analysis, how many fields analyzed, blinded or not?
Response 10: We apologize for failing to clearly describe the experimental methodology. For histopathological analyses, we analyzed three samples per group using a blinded method. According to your comments, we have made additions to lines 609-610 of the revised draft and highlighted them in red.
Comments 11: Figure S5 needs more explicit labeling—please clearly mark which bands correspond to which proteins and specify molecular weights clearly for transparency. You have provided original blots for reference; here you can clearly label the bands. Also, the legend needs to be more explanatory. It is difficult to understand why 3 images for each? Are they replicates (n=3)?
Response 11:Thank you for your professional comments on this study.Based on your comments, we have clearly labeled the corresponding bands and molecular weights of the proteins in Figure S5. The legend in Figure S5 has been changed to “Original protein bands for NLRP3, Caspase1, ASC and GAPDH (n=3)”, while each band has 3 images, representing three repetitions.Following your comments, we have modified Figure S5 in the Supporting Materials. However, it will not affect the experimental results of this study.
Comments 12: Why certain bacterial genera were selected for specific analysis? Are these relevant to influenza-induced inflammation? I do not understand here the objective. You have to clarify this.
Response 12: Thank you for your valuable suggestions on our research. We have clarified the association of conditioned pathogens (Vibrio, Pseudoalteromonas, Agathobacter, Erysipelatoclostridium) and probiotics (Akkermansia, Dialister, Faecalibacterium,cKlebsiella) in influenza-induced inflammation and lipid metabolism abnormalities in detail in lines 477-489 of the revised manuscript and labeled them in a red font.
Comments 13: Explain if lipid metabolism changes observed (particularly cholesterol and TG levels) directly correlate with microbiota changes, as implied by correlation analysis, or whether there are other potential intermediate mechanisms? You can include this in discussion.
Response 13: Thank you for your valuable comments on this study. Based on your comments, we have clarified lines 496-500 in the discussion and highlighted them in red.
Comments 14: You have mentioned your few limitations in the discussion section. It’s better either to add in conclusion or have separate heading “Limitations of the study”. You have not addressed the potential direct interactions between aromatic molecules and host or viral proteins.
Response 14: Thank you for your valuable comments on this study. Following your comments, we have added the limitations of this study and future research in lines 691-698 of the revised manuscript and highlighted them in red font.
Comments 15: You have to address data presentation errors such as repeated "Error! Reference source not found" references. Most of the places this is appearing. You have to correct it.
Response 15: We apologize for failing to use the correct reference formatting in the previously submitted manuscript. In this revised manuscript (Word version), we have checked that all references are formatted in accordance with the requirements of the journal. This can be seen in line 732 of the revised manuscript under “References”.
Comments 16: No standardisation of the figure legends, without clear experimental conditions and number of replicates. Poor and non-explanatory figure legends.
Response 16: We apologize for the unclear description of the figure legends. Based on your comments, we have added experimental conditions to the corresponding experimental steps and added the number of repetitions to the legend in the revised manuscript and Supplementary Materials, and labeled them in red font.
Comments 17: "The therapeutic effect of AC was verified by evaluating the antiviral effect of AC on model mice, and the improvement of lung and colon inflammatory injury, oxidative stress and NLRP3 inflammasome."
It’s better to revise the sentence to:
"The therapeutic effect of AC was verified by evaluating antiviral efficacy in mouse models, including improvements in lung and colon inflammation, oxidative stress, and suppression of the NLRP3 inflammasome."
Response 17: Thank you very much for your positive suggestions for this study. According to your comments, we have replaced the sentence in lines 20-22 of the revised draft and highlighted it in red.
Significant structural and grammar issue. Please rectify. Some example below:
Comments 1:. "Our team screened four volatile compounds from aromatic Chinese medicines and defined the concomitant application of the four compounds as aromatic molecular compatibility (AC) through molecular compatibility theory, but the therapeutic effect and mechanism of AC were unclear."
It’s better to revise the sentence to:
"Our team previously identified four volatile compounds from aromatic Chinese medicines. Based on molecular compatibility theory, we defined their combination as aromatic molecular compatibility (AC), though its therapeutic effects and underlying mechanisms remain unclear."
Response 1: Thank you for your positive suggestions about the structure and language of this manuscript. According to your comments, we have replaced the sentence in lines 15-18 of the revised draft and highlighted it in red.
Comments 2: "Given the significant morbidity and mortality associated with influenza pandemics each year and the potential for influenza to have a similar impact as that of COVID-19..."
Add comma after "each year”
Response 2: Thank you for your positive suggestions about the structure and language of this manuscript. According to your comments, We have replaced “each year and the potential” with “each year , and the potential” on lines 44-45 of the revised draft and in red.
Comments 3: The phrase "Error! Reference source not found." frequently appears, indicating citation or referencing issues throughout the manuscript.
Response 3: We apologize for failing to use the correct reference formatting in the previously submitted manuscript. In this revised manuscript (Word version), we have checked that all references are formatted in accordance with the requirements of the journal. This can be seen in line 732 of the revised manuscript under “References”.
Comments 4: "This study employed a mouse model of influenza virus infection induced by administration of nasal drops of the IAV……..
It’s better to revise the sentence to:
"This study employed a mouse model infected intranasally with Influenza A virus (IAV) PR8 strain (10³ TCIDâ‚…â‚€, 50 μL)."
Response 4: Thank you for your positive suggestions about the structure and language of this manuscript. According to your comments, we have replaced the sentence in lines 139-140 of the revised draft and highlighted it in red.
Comments 5: "mortality commenced on the sixth day of treatment in the vehicle group compared with the control group and persisted until the ninth day."
It’s better to revise the sentence to:
"Mortality began on day 6 in the vehicle-treated group compared with controls and continued until day 9 post-infection."
Response 5: Thank you for your positive suggestions about the structure and language of this manuscript. According to your comments, we have replaced the sentence in lines 142-143 of the revised draft and highlighted it in red.
Comments 6: Fig. 3 legend: "Representative images of histologic lesions in the lungs. Amelioration of histologic lung lesions by treatment with AC."
It’s better to revise the sentence to:
"Representative images showing histopathological changes in lung tissues and their amelioration following AC treatment."
Response 6: Thank you for your positive suggestions about the structure and language of this manuscript. According to your comments, we have replaced the sentence in lines 196-198 of the revised draft and highlighted it in red.
Comments 7: There are many language issues in the manuscript. Authors have to take professional English editing service.
Response 7: Thank you for your comments on this manuscript. Based on your comments, we have revised the language of the revised draft with the help of AJE. We have acknowledged them in 728 line of the revised manuscript and highlighted them in red.
Response 8: In section “Animal Modeling and Treatment”: "The mouse model of influenza virus infection has been described in the literature."
Reference is needed here
Response 8: Thank you for your careful reading of this study, due to our negligence in not citing this section in the literature. Following your comments, We have cited the corresponding Reference in this section of the revised manuscript and highlighted it in red (line 591).
Response 9: Verify all abbreviations used in figures/tables, and are clearly explained within figure legends or table notes. Majority of the abbreviations used in figures and tables are not mentioned in the abbreviation list.
Response 9: Thank you for your valuable comments to this study. Based on your comments, we have carefully verified the abbreviations used in the article and added to the “Abbreviations”, specifically in line 730 in the revised manuscript and marked in red font.
Response 10: There is a noticeable repetition of citations referencing previous work using "Error! Reference source not found," making it unclear exactly which references correspond to the authors' previous work. This error you have to correct to confirm if you are mentioning any othere reference beyond ref no. 41.
Response 10: We apologize for failing to use the correct reference formatting in the previously submitted manuscript. In this revised manuscript (Word version), we have checked that all references are formatted in accordance with the requirements of the journal. As can be seen in line 98 of the revised manuscript, only Reference No. 41 was cited for the previous study.
Round 2
Reviewer 2 Report
Comments and Suggestions for Authors
Manuscript is significantly improved by the authors, and now can be accepted in its current form.